# A qualitative study on factors influencing health workers' uptake of a pilot surgical antibiotic prophylaxis stewardship programme in selected Georgian hospitals

Sideeka Narayan[1], Sahil Khan Warsi[1]*, Iago Kachkachishvili[2], Osiko Kontselidze[2], Mariam Jibuti[2], Nino Esebua[2], Ana Papiashvili[2], Danilo Lo Fo Wong[1], Ketevan Kandelaki[1]

**1** World Health Organization (WHO) Regional Office for Europe, Copenhagen, Denmark, **2** Institute of Social Studies and Analysis, Tbilisi, Georgia

* warsis@who.int

## Abstract

Antimicrobial misuse in surgical antibiotic prophylaxis (SAP) can include the inappropriate use of broad-spectrum antibiotics or prolonged dosing. In 2021, a pilot antimicrobial stewardship programme (ASP) was launched in Georgia, which involved developing and adapting SAP guidelines, establishing an interprofessional SAP prescribing approach, collecting surgical site infection (SSI) data via routinely collected data and telephonic patient follow-ups, and providing surgical unit staff with prescribing feedback and training on antimicrobial resistance (AMR) and antimicrobial stewardship (AMS). ASP introduction was staggered across ten hospitals over three years. This study explored behavioural determinants of surgical teams' ASP uptake in five hospitals where the ASP was introduced or about to be introduced. Findings primarily concerned epidemiologists' and nurses' ASP-related behaviour. Those at ASP non-introduced hospitals were less involved in the SAP prescribing process, had lower AMR awareness, and lacked professional development opportunities. Those at ASP-introduced hospitals exhibited higher AMR knowledge and felt ASP participation boosted confidence, facilitated work, and furnished key professional development. Results indicate interprofessional collaboration on SAP prescribing supported ASP uptake across teams, and investment in health worker training and administrative encouragement ensured effective ASP participation and implementation. Findings highlight the crucial role of epidemiologists in SAP and illustrate a need for developing Georgian nurses' AMR competencies as a vehicle to address public AMR knowledge gaps. Longer-term ASP uptake will need to consider the regulatory context in which hospitals lack access to national-level SSI data and feedback on SSI reporting but are fined for reporting non-compliance. Despite resource limitations and a small sample size, the study engaged all pilot ASP health workers. Respondents' inexperience of qualitative research participation and ensuant hesitation limited exploration of motivational factors supporting health workers' ASP uptake, which could be explored in further research.

**Data availability statement:** All relevant data are within the paper and its Supporting Information files.

**Funding:** This study was funded by the WHO Control of Antimicrobial Resistance Programme, under the Division of Communicable Diseases, Environment and Health at the WHO Regional Office for Europe using voluntary contributions of the Ministries of Health of Germany and the Kingdom of the Netherlands. The funders had no role in study design, data collection and analysis, decision to publish, or preparation of the manuscript. Funding covered all development and implementation costs of the study, including partial staff costs of all authors for technical support provided.

**Competing interests:** The authors have declared that no competing interests exist.

## Introduction

Antimicrobial resistance (AMR) is a global threat to public health. AMR occurs when micro-organisms, such as bacteria, viruses, parasites and fungi, change so that they are no longer affected by antimicrobial medicines used to treat them. The development and spread of AMR is accelerated by the inappropriate use of antimicrobials, resulting in harder-to-treat infections [1–3]. Bacterial AMR in the WHO European Region was associated with over half a million deaths in 2019, putting considerable pressure on hospitals. Annually, AMR-related infection treatments in the European Union (EU) and European Economic Area (EEA) cost patients over 9.5 million extra days in hospital, and higher health expenditure and reduced workforce productivity cost countries approximately €11.7 billion [4–6]. 13The misuse of antimicrobials in health-care settings is one of the key modifiable drivers of the emergence of AMR. This issue deserves particular attention in surgical wards where antibiotic prophylaxis is routinely administered prior to surgery to help decrease the risk of postoperative infections. Optimizing surgical antibiotic prophylaxis (SAP) requires an appreciation and understanding of the behavioural and cultural context influencing health-care professionals' practices and decisions.

Antimicrobial stewardship programmes (ASPs) are one of the most cost-effective interventions to optimize antimicrobial use, improve patient outcomes, and reduce the development and spread of AMR [7,8]. It involves a systematic approach to educate and support health-care professionals to follow evidence-based guidelines for prescribing and administering antimicrobials. Educating the health workforce is crucial, as they are key in safeguarding antimicrobial effectiveness. Successful ASPs not only equip practitioners with information, but do so by attending to the behavioural factors affecting programme uptake [9,10].

In Georgia, as in other countries, the inappropriate use of antibiotics, including for surgical prophylaxis, is characterized by a high use of broad-spectrum antibiotics and prolonged dosing, as evidenced in a 2020 SAP assessment across several hospitals in Tbilisi [11–14]. Evidence shows that such inappropriate SAP practices can have serious consequence for patients undergoing surgical procedure, including increased morbidity and mortality [15]. Actions to contain the development and spread of AMR in Georgia have thus far mainly focused on strengthening AMR surveillance systems and implementing modern methods for infection prevention and control (IPC). However, limited actions have been implemented to establish ASPs and understand the factors affecting their uptake among health professionals [16,17].

In 2021, as part of Georgia's 2017–2020 National action plan (NAP) to contain the spread of AMR, the International Centre for Antimicrobial Resistance Solutions (ICARS) began a three-year ASP to develop and introduce SAP guidelines in ten hospitals, aiming for 60% hospital guideline compliance within 12 months from ASP introduction. The ASP involved AMR and antimicrobial stewardship (AMS) training for surgical unit staff; data collection on surgical site infection (SSI) rates via post-surgical patient telephone interviews, and on antibiotic prescribing via monthly point prevalence surveys (PPS); and feedback to surgical teams to adjust prescribing practice. Regarding health worker behaviours, the ASP primarily involved surgeons following new SAP guidelines, epidemiologists conducting PPS, and nurses conducting post-surgical patient interviews. The Ministry of Internally Displaced Persons from the Occupied Territories, Labor, Health and Social Affairs of Georgia (MoIDPLHSA) and ICARS signed a memorandum of understanding in September 2021, followed by an official project launch where ASP participating staff at all ten hospitals were informed on ASP objectives and timelines and their specific roles. ASP introduction was planned to begin in 2022 in overlapping stages for three hospitals in year 1, four in year 2, and three in year 3.

In collaboration with the MoIDPLHSA and National Centre for Disease Control and Public Health of Georgia (NCDC), the WHO Regional Office for Europe and WHO Country

Office in Georgia undertook a study in 2022 to identify barriers to and enablers for implementing the ICARS ASP during the first year of its introduction. The study was conducted in parallel to the ICARS ASP and followed the WHO Regional Office for Europe Tailoring Antimicrobial Resistance Programmes (TAP) method to identify barriers to and drivers of behaviours contributing to ASP uptake. National researchers from the Institute of Social Studies and Analysis (ISSA) conducted fieldwork across selected participating hospitals to understand factors affecting health workers' ASP-related behaviours.

## Methods

### Ethics statement

The study was conducted in accordance with the Helsinki Declaration (1964, revised 2013), and received ethical approval from the WHO Research Ethics Committee, protocol number ERC.0003668, and the Institutional Review Board of the National Centre for Disease Control and Public Health of Georgia, IRB number 2021–066. All participants received information on research ahead of recruitment; were given the chance to ask questions before, during, or after their participation; and were free to cease participation in the study or refuse to answer any questions at any time. All participants provided written and verbal consent to participate.

### Study design and setting

Data collection and analysis were conducted using the capability, opportunity, and motivation for behaviour change (COM-B) theory, which is part of the behaviour change wheel (BCW) framework to understand and evaluate health-related behaviour interventions [18]. To ensure a rigorous, theory-based approach to intervention development, the TAP method employs the BCW framework, which links specific sources of behaviour with discrete intervention functions and policy categories based on research evidence. Using this framework, the TAP method ensures developed interventions address relevant behavioural barriers and drivers with evidence-based approaches. The COM-B theory at the core of the BCW framework is built on a review of 19 existing behaviour change frameworks, each of which approaches public health behaviour from various perspectives, and presents a model for exploring overlapping and interacting individual and contextual factors that affect individuals' performance of public health behaviours. The COM-B theory holds that public health behaviour change is influenced by the interlinked factors of individuals' capability, opportunity, and motivation to enact specific behaviours. Fig 1 below from the TAP Quick Guide illustrates how the COM-B factors are approached through the TAP process [19].

Data was collected in 2022 from five of the ten hospitals. The research sites represented regional and hospital variation. Three hospitals were selected from Tbilisi and two from the regions, representing a large private, smaller private, teaching, maternity, and military hospital.

### Research participants

The primary target groups were the ICARS ASP team at each hospital: one hospital administrator, one AMR champion, one epidemiologist, one surgical unit chief nurse and surgeons. In two hospitals, the hospital administrator or AMR champion was also a surgeon. In such cases, the individual participated in research in their administrator or AMR champion capacity and additional questions were also asked relating to their role as a surgeon. Research was also conducted with secondary target groups affecting the behaviour of primary target group participants. This included a clinical pharmacist, health professional association representatives, and pharmaceutical company representatives. Primary target group participants were sampled

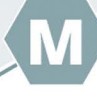

**Capability**

- **Knowledge**: Practitioner and consumer knowledge of ABR and AMR
- **Efficacy**: Individual belief of practitioner or consumer that they have the capability to influence AMR and ABR
- **Skills**: Practitioner skills, and trust in their own skills, including interpersonal communications with consumer on prescribing practices
- **Resilience**: Practitioner stamina and willpower to follow-though on intentions to address AMR
- **Understanding**: Individual understanding of how to reduce the spread of AMR in a hospital, clinic or other setting, or understanding of the individual role in reducing AMR and its implications

**Opportunity**

- **Access**: Practitioner access to infection control supplies, such as soap and hand gel; consumer access to information, such as knowledge materials; consumer access to safe and quality medicines and vaccines; convenience of antibiotic consumption practices for consumers
- **Regulations**: Implementation of infection control interventions; implementation of antimicrobial stewardship practices
- **Culture**: Open culture to report health care-associated infections; a 'hygiene' culture; social practices around the use of traditional vs 'western' medicines
- **Social norms and values**: Peer pressure for practitioners to influence or adapt infection control practices

**B** PRACTITIONERS ANTIBIOTIC PRESCRIBING PRACTICES AND BEHAVOUR

**Motivation**

- **Beliefs**: Practitioner belief in prudent prescribing practices; consumer belief in practitioner recommendations on the safe use of antibiotics
- **Values**: Practitioner individual values for prescribing habits; individual values for patient safety and service delivery quality
- **Intentions**: Practitioner motivation to improve prescribing practices and learn more about AMR; interpersonal communications between the practitioner and consumer impacting how the practitioner's recommendations are received
- **Inhibitions**: Factors that may affect the practitioner in decision-making for antibiotic prescription practices

**Fig 1. The COM-B model adapted to AMR.** The COM-B theory was employed to understand the behaviour of each target group outlined below in relation to their function under or related to the ICARS ASP. Across the target groups, the behaviours explored included SAP prescribing and adherence to guidelines, conducting ASP-related surveillance and feedback, and integrating the ASP in hospital and surgical team practice. The behaviours explored for each target group are presented in Table 1 in the results section below.

to include the entire ASP team at each hospital, and secondary target group participants were purposively sampled based on their availability and accessibility. Participants were recruited into the study from 1 May to 1 July 2022.

### Data collection and analysis

Only one participant in each target group was engaged in the ASP at each hospital, which was the entire surgical team at most hospitals, thus data was almost entirely collected via semi-structured, in-depth interviews (IDIs). Only one focus group discussion (FGD) was conducted with surgeons from the three Tbilisi hospitals. IDIs and the FGD were conducted using research guides, with questions developed via a COM-B-inspired activity organized by the WHO Regional Office for Europe with the participation of representatives from ICARS, NCDC, MoIDPLHSA, and the ASP hospitals. Questions addressed individuals' knowledge and perceptions of AMR and SAP, views on the ASP, and factors affecting ASP incorporation into hospital practice. The guides were initially developed in English and then translated to Georgian.

Prior to data collection, the WHO Regional Office for Europe ICARS and NCDC researchers conducted a week-long workshop with the national research team. During this workshop, national researchers were informed about the Georgian context of SAP and SSIs, the health system context of Georgia, and trained on COM-B theory and its use in research, data collection, and analysis. National researchers, jointly with WHO Regional Office for Europe ICARS and NCDC representatives, reviewed each research guide, line-by-line, for linguistic and contextual appropriateness, and simultaneously adapted Georgian- and English-language versions to ensure coherence across all guides. The guides were then piloted before research to ensure feasibility.

All discussions were audio-recorded and transcribed in Georgian before analysis. Transcripts were anonymized using participant IDs, ensuring no identifying details were included. As WHO Regional Office for Europe researchers could not be present during the initial stage of data collection, four initial transcripts were translated in English and shared with them for feedback and for preliminary analysis training. A coding framework was developed based on sections of the research guides. National and WHO Regional Office for Europe researchers coded two initial transcripts separately and then compared results to identify differences. Based on these results, feedback was provided to national researchers and the coding framework was adapted to allow for emerging themes across transcripts. A second round of analysis was conducted with the remaining two English transcripts, with almost identical coding by all researchers. After this exercise, all transcripts were then coded and analysed directly from Georgian by national researchers.

Coded data, observations, and quotations were organized into a Microsoft Excel spreadsheet for each target group in Georgian. National researchers developed a thematic report based on analysed data which was used by WHO Regional Office for Europe researchers to conduct a COM-B analysis of research findings by target group and hospital type.

## Results

A total of 26 IDIs and one FGD were conducted with 31 participants from five hospitals. At the time of data collection, the ASP had been introduced in three of the five hospitals. Findings on factors affecting ASP uptake show similarities across primary target groups as well as differences between ASP-introduced and non-introduced hospitals. Secondary target group participants' responses mainly captured factors affecting primary target groups' behaviours, and were thus included in primary target group COM-B analysis. An overview of participants by target group, ASP-related behaviours and research activities are presented in Table 1.

Barriers to ASP uptake in non-introduced hospitals mostly related to nurses' and epidemiologists' behaviours, as discussed below, though some were shared across target groups. Table 2 summarizes these barriers by COM-B factor.

**Table 1. Participants by target group, ASP-related behaviour, and research activity.**

| Target group | ASP-related behaviour | Research activity | Total participants |
|---|---|---|---|
| *Primary target group (ASP participants)* | | | |
| Hospital administrator | Leads adoption and dissemination of SAP guidelines and ASP introduction | IDI | 4 |
| Hospital AMR champion | Coordinates ASP introduction, data collection/analysis and feedback | IDI | 3 |
| Surgeons | Receive training and use developed SAP guidelines | FGD | 5 |
| | | IDI | 5 |
| Epidemiologists | Collect and analyse SSI/PPS data for SAP ASP | IDI | 5 |
| Study nurse | Collects SSI data from post-surgery patients | IDI | 5 |
| *Secondary target group (influencers)* | | | |
| Hospital pharmacists | Influence antibiotic availability and prescribing process | IDI | 1 |
| Professional association representatives | Influence behaviour of primary target group professionals | IDI | 2 |
| Pharmaceutical company representatives | Directly advertise and provide antibiotics to hospitals | IDI | 1 |
| **Total** | | | **31** |

**Table 2. Barriers to AMS-related behaviours by COM-B factor.**

| COM-B Factor | Barriers to AMS-Related behaviours |
|---|---|
| **Capability** | **Awareness gaps**: Participants were unaware of hospital ASP participation or their roles. Epidemiologists perceive low facility SSI incidence contrary to actual incidence. **Skills gaps**: Health workers can have skills gaps in effective communication on antibiotic use. Some administrators/surgeons don't follow interprofessional collaborative approach to patient antibiotic treatment. **Knowledge gaps**: Nurses and Epidemiologists indicated AMR knowledge gaps, lack up-to-date information, or unfamiliar with AWaRE classification. |
| **Opportunity** | **Professional development**: Epidemiologists and nurses lacked AMR training opportunities **Data availability**: Epidemiologists did not have access to national-level SSIs data. **Reporting regulations:** Hospitals (including ASP-introduced) do not receive feedback on SSI reporting, but fined for any non-compliance identified in reporting. **Public awareness**: In context of AMR unawareness, antibiotics demand, and OTC antibiotic availability patients can ignoring doctors' advice on antibiotics |
| **Motivation** | **ASP Hesitancy**: health work workload, effectiveness, patient acceptance, and other factors |

## Behaviour

ASP-introduced hospital participants reported active involvement in ASP activities, while ASP non-introduced hospitals' participants had limited involvement in similar activities. Variation was primarily observed across hospitals in epidemiologist and nurse involvement in SAP. Epidemiologists at all hospitals reported conducting surveillance on health-care associated infections (HAIs), trainings on issues such as hand hygiene and disinfection-sterilization or HAI reporting to NCDC. ASP-introduced hospital epidemiologists supervised medical staff on and monitored antibiotic prescription for guideline compliance. Conversely, ASP-non-introduced hospital epidemiologists were not involved in antibiotic selection or the prescribing process but believed they should be.

> *"As for monitoring the use of antibiotics, I do not monitor what antibiotic the doctor pre-scribed and how. We talked about this issue with the quality service, that it is better if we get more actively involved in the issue. We have a plan to take one in ten cases and check if antibiotics have been prescribed according to the guideline."*

> *– Epidemiologist, Hospital 1, ASP non-introduced*

Nurses in Georgia administer prescribed antibiotics and supervise therapy; they do not have the right to prescribe. ASP-introduced hospital nurses were more involved in patients' treatment, regularly communicating with them and providing information on AMR and the ASP. They interviewed patients one-week and one-month post-surgery, recording information on prescription and non-prescription antibiotic use, complaints and complications. ASP non-introduced hospital nurses were not involved in such activities.

Significant behavioural differences across hospitals were not observed for other target groups. Surgeons develop patients' antibiotic treatment plans in consultation with other specialties, although one ASP non-introduced hospital administrator indicated some staff might rely on experience rather than evidence-based guidelines. Administrators, in general, ensure the quality and safety of patient care and implement treatment guidelines and monitor their compliance. The AMR champion role was created for ASP; the majority of those in this role were already engaged in some form of administration or treatment strategizing.

## Capability

Although all participants were informed via the 2021 ASP launch, most ASP non-introduced hospital participants were unaware of their hospital's participation and their roles. Differences in knowledge, skills and training were specifically observed among nurses and epidemiologists. Surgeons stated nurses lack required ASP competencies. A professional association representative reported AMR knowledge gaps among epidemiologists, citing insufficient up-to-date information and confirmed by ASP non-introduced hospital epidemiologists' unfamiliarity with the Access, Watch, Reserve (AWaRe) classification. ASP non-introduced nurses and epidemiologists expressed an interest to participate in trainings to strengthen knowledge of AMR and the ASP.

ASP-introduced hospital epidemiologists and nurses felt confident in their skills and reported significant improvement as a result of ASP trainings. Nurses expressed increased AMR and SAP knowledge, which facilitated patient communication, and believed the additional ASP duties and understanding of AMR were interesting and important for professional development. Administrators added that extra communication was often required for patients and their families to encourage appropriate antibiotic use. Epidemiologists demonstrated awareness of the AWaRe classification and proactive use of guidelines, reporting this as an important tool for effective monitoring and supportive to their work.

> *"There were several cases when the doctor did not prescribe an antibiotic, but the patient replied that they still take it on their own. They even had complaints about it. They still think it is impossible for the treatment to go well without antibiotics."*

> *- Nurse, Hospital 5, ASP-introduced*

A few capability findings related to participants' perceptions. Regarding SAP guideline application, some ASP non-introduced hospital administrators and surgeons believed a consultative approach to patient antibiotic treatment planning was not necessary as national guidelines outlined available options. Epidemiologists perceived their facility's SSI incidence was low at one to two cases per year.

> *"We don't need to consult with colleagues [such as infectious disease specialists] on which antibiotic to use or replace. The Ministry has given us [SAP Guidelines] with the names of only two antibiotics."*

> *-Surgeon, Hospital 1, ASP non-introduced*

## Opportunity

ASP-introduced hospital participants reported that the hospital-specific adaptation of national SAP guidelines and introduction of the digital patient recording system facilitated antibiotic treatment and planning. Several ASP-introduced hospital participants mentioned an initial increase in workload following ASP introduction but reported that this was integrated into routine practice.

> "…protocols developed will help health-care facilities use antibiotics more rationally. Besides, we, as doctors, will obtain knowledge and will disseminate it within the facility... Some surgeons are employed at several [other] hospitals and will disseminate the knowledge there as well."

> -AMR champion, Hospital 4, ASP-introduced

While epidemiologists across all hospitals mentioned having continuous medical education opportunities on various topics, ASP non-introduced hospital epidemiologists and nurses reported not receiving AMR-specific training. AMR champions at ASP-introduced hospitals further reported ASP communication was presented to all services, and that SAP conferences and trainings were open to all staff, broadening staff involvement and exposure to the ASP. One epidemiologist said ASP scale-up and sustainability would depend on government enforcement through integration in national health plans.

Opportunity barriers reported by epidemiologists related to reporting on SSIs. They do not have access to national-level information on SSIs. Additionally, while hospitals do not receive feedback from NCDC on SSI reporting, they are fined for non-compliance with reporting regulations.

> "In case of HAIs we report what went wrong and why HAI developed. We have no feedback but, in case of non-compliance with regulations, the clinic might be fined."

> - Epidemiologist, Hospital 5, ASP-introduced

Participants cited the lack of public AMR awareness and demand for antibiotics as a social opportunity barrier. They added that patients could deviate from doctors' advice by buying over-the-counter (OTC) antibiotics. Relatedly, an AMR champion emphasized treatment plans must be effectively communicated to patients to increase treatment compliance and confidence. ASP-introduced hospital nurses further reported that patients positively responded to follow-up calls providing information on AMR, with only few becoming suspicious or anxious about their health.

> "[One challenge] is that the patient or their family member demands antibiotics… we convince them that it's not needed. We inform them about the possible side-effects of inappropriate use of antibiotics… The majority of patients are satisfied when you call them after the surgery to check on them … in rare cases… even though they are informed in advance to expect two calls, they can become suspicious."

> -Nurse, Hospital 1, ASP-introduced

> "If the information is communicated correctly to the patients and the results are also favourable, the patient will go along with you. I think it is always possible to overcome stereotypes if you try."

> - AMR champion, Hospital 4, ASP-introduced

## Motivation

Participants across target groups reported initial hesitancy to the ASP, related to workload, effectiveness, patient acceptance, and other factors. However, they indicated that hospital context influenced the willingness and ability to support and participate in the ASP. ASP-introduced hospital administrators reported the programme strengthened staff knowledge and responsibility. Nurses and epidemiologists specifically welcomed such qualification-raising opportunities and believed they are integral to ASP implementation. Epidemiologists also believed that administrators play a role in encouraging ASP acceptance and adherence.

> *"The initial attitude was skeptical. However, it was seen that the [SAP] was successful… This requirement has been strengthened by the guidelines, and our administration actively supervises us."*
>
> *- Epidemiologist, Hospital 3, ASP-introduced*
>
> *"….it would be better if we get involved, receive information and solve the problem together."*
>
> *-Head Nurse, Hospital 4, ASP-introduced*
>
> *"We have had good results since the beginning. Our fears did not come true and even one dose proved to be sufficient for preventive purposes. These are the results we have had so far."*
>
> *- Surgeon, Hospital 1, ASP-introduced*

## Discussion

The findings presented above highlight drivers supporting the ASP uptake in hospitals where it was introduced, and a few barriers potentially affecting longer-term ASP scale-up and sustainability. Findings primarily relate to epidemiologists' and nurses' ASP behaviours, perhaps in part because other target groups' ASP roles did not differ significantly from their routine functions.

Firstly, participants across target groups appreciated the new dynamic in SAP prescribing introduced under the ASP, which encourages involvement of an interprofessional team. Broader research on SAP ASPs emphasizes how successful ASP implementation requires overcoming paternalism in decision-making and fostering interprofessional consultation [20–22]. Unlike counterparts at other hospitals, epidemiologists and nurses from ASP-introduced hospitals were respectively involved in SAP treatment planning and in patient follow-up. Research participants from both these groups valued the increased educational opportunities and concomitant professional development offered under the ASP, despite an indication of having increased workloads, and felt it successfully supported them in carrying out their roles. Their view echoes research on SAP ASPs that stresses the need to invest in health worker education to ensure their effective participation and consequently successful ASP implementation [23–25].

Research also showed epidemiologists and nurses across all hospitals play an important role in SAP. ASP-participating hospital participants not only expressed this belief, but also explained ASP participation boosted their professional confidence. Nurses in particular mentioned this fact with regard to discussing AMR with patients. Global research illustrates epidemiologists are critical to ASP efforts in leadership support, sharing surveillance data or

outbreak alerts, bridging gaps between departments, and treatment algorithm development [24,26]. Nurses, in turn, have been shown to be crucial across the health economy, especially in fostering health literacy of AMR through effective communication and education [23,26]. Investment in developing nurses' AMR competencies could be particularly important in the Georgian context. As surgeon and administrator participants noted, while nurses played an important role in ASP, many did not have sufficient qualifications. This is corroborated by reporting that patient and health outcomes in Georgia are affected by a nursing shortage, affected by factors such as low wages, professional development opportunities, labour migration, and social stigma [27–29].

Barriers to health worker ASP uptake were connected to the ASP delivery and health system context. Non-introduced hospital participants lacked awareness of the SAP project. This could be due to several factors, including staff turnover and limited communication on the ASP with the hospitals between the programme launch and introduction. Regarding health system context, the unavailability of national-level information on SSIs, combined with fines for not reporting and a lack of feedback to hospitals on SSI reporting, could impede longer-term reporting behaviour and limit access to up-to-date information to support appropriate SAP. While behaviour change interventions are effective to address AMR, wider health system level changes, such as effective SSI reporting and feedback, are necessary to support ASP scale-up and sustainability [30,31].

Another factor cited as affecting long-term ASP sustainability was patients' demand for and access to antibiotics OTC, leading to their disregard of doctors' treatment advice. OTC antibiotics demand is affected by multiple factors including AMR or antibiotics knowledge gaps, individual convenience and cost considerations, health provider AMR awareness, and policy and guideline enforcement [32,33]. Research in Georgia, as in other countries in the region, shows OTC antibiotics demand is linked to patient convenience in wanting to save time and money on doctor appointments or buying drugs from more expensive pharmacies rather than cheaper community pharmacies, as well as knowledge gaps leading to over confidence in self-prescribing [34,35]. Public awareness of AMR is a widespread issue in the country, and study nurses did report their ASP training equipped them with knowledge and skills to effectively communicate with patients on AMR and appropriate antibiotic use, which administrators indicated was a particular challenge. In this way, the ASP could address an important communication skills gap. Investing in training health workers is important. Public attitudes towards AMR in Georgia and elsewhere have been shown to be effectively addressed through education via schools or by health workers, whom the public see as trusted sources of health information, rather than solely via information campaigns [11,36–38].

Two challenges limiting study results related to the time available for research and to navigating health worker concerns around participation. Resource constraints decreased the available time for research and analysis. To address this, time was allocated for researcher training on SAP and ASP, theoretical approaches for data collection and interpretation, conducting research and analysis, and collaborating with diverse research participants. Health workers, especially nurses, were new to participating in qualitative research, and despite providing informed consent, were concerned of being professionally assessed. This fact limited participants' responses and, subsequently, information on motivation factors affecting participants' behaviours. Nonetheless, almost all ASP participants were included in research, and findings provide relevant insights following other qualitative studies revealing SAP ASPs as affected by multiple factors. These include missed opportunities due to professional hierarchies, low health worker competencies or confidence, and unawareness of local AMR epidemiology [20,39,40].

## Conclusion

Study findings indicate the introduction of interprofessional collaboration on SAP supported ASP uptake among all participants. Epidemiologists' and nurses' uptake was encouraged by the additional professional development opportunities provided under ASP and facilitated by increased SAP and AMR knowledge. Further inquiry should explore motivation factors supporting health workers' ASP uptake, which would require addressing health worker concerns on research participation.

## Supporting information

**S1 Checklist. Inclusivity in global research.**
(DOCX)

**S1 Data. COM-B findings for thematic analysis.**
(DOCX)

## Acknowledgments

The authors would like to thank Marika Tsereteli (ICARS) for support and coordination on in-country research; David Chakhunashvili (NCDC) for support in researcher training; Marine Baidauri and colleagues at the MoIDPLHSA for clarifying questions arising from research; Ute Wolff Sönksen (Statens Serum Institut Denmark) for support in initial planning of the research; and Giorgi Kurtsikashvili (WHO Country Office in Georgia) and the leadership of the NCDC and MoIDPLHSA for coordination and support to the project. Additionally, we would like to thank Paul Csagoly for his contributions to the editing of the manuscript.

The authors affiliated with the World Health Organization (WHO) are alone responsible for the views expressed in this publication and they do not necessarily represent the decisions or policies of the WHO.

## Author contributions

**Conceptualization:** Sahil Khan Warsi, Ketevan Kandelaki.

**Data curation:** Iago Kachkachishvili, Osiko Kontselidze, Mariam Jibuti, Nino Esebua, Ana Papiashvili.

**Formal analysis:** Sahil Khan Warsi, Iago Kachkachishvili, Osiko Kontselidze, Mariam Jibuti, Nino Esebua, Ana Papiashvili, Ketevan Kandelaki.

**Methodology:** Sahil Khan Warsi, Iago Kachkachishvili, Ketevan Kandelaki.

**Project administration:** Danilo Lo Fo Wong.

**Resources:** Sahil Khan Warsi, Iago Kachkachishvili, Ketevan Kandelaki.

**Supervision:** Sahil Khan Warsi, Danilo Lo Fo Wong, Ketevan Kandelaki.

**Visualization:** Iago Kachkachishvili, Osiko Kontselidze, Mariam Jibuti, Nino Esebua, Ana Papiashvili.

**Writing – original draft:** Sideeka Narayan, Sahil Khan Warsi, Ketevan Kandelaki.

**Writing – review & editing:** Sideeka Narayan, Sahil Khan Warsi, Ketevan Kandelaki.

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
