## [Decision Letter · Decision Letter 0]

2 Dec 2024

PGPH-D-24-01470

A qualitative study on factors influencing health workers’ uptake of a pilot surgical antibiotic prophylaxis stewardship programme in selected Georgian hospitals

Dear Dr. Sahil Warsi,

Thank you for submitting your manuscript to PLOS Global Public Health. After careful consideration, we feel that it has merit but does not fully meet PLOS Global Public Health’s publication criteria as it currently stands. Therefore, we invite you to submit a revised version of the manuscript that addresses the points raised during the review process.

We look forward to receiving your revised manuscript.

Kind regards,

Vrinda Nampoothiri

Academic Editor

Journal Requirements:

2. Your current Financial Disclosure states, “The author(s) declare financial support was received for the research, authorship, and/or publication of this article. This study was funded by grants provided by the Ministries of Health of Germany and the Kingdom of the Netherlands, which had no role in data collection, analysis, or interpretation of data. The WHO Control of Antimicrobial Resistance Programme, under the Division of Communicable Diseases, Environnement and Health, was the recipient of these grants and coordinated the development of this study. This study was supported by the WHO Regional Office for Europe.

The funders had no role in study design, data collection and analysis, decision to publish, or preparation of the manuscript.”.

However, your funding information on the submission form has no information. Please indicate by return email the full and correct funding information for your study and confirm the order in which funding contributions should appear. Please be sure to indicate whether the funders played any role in the study design, data collection and analysis, decision to publish, or preparation of the manuscript.

3. In the online submission form, you indicated that "Requests for the data used and/or analysed during the current study should be directed to WHO Regional Office for Europe.". 

3. Uploaded as supplementary information.

4. Please provide an Author Summary. This should appear in your manuscript between the Abstract (if applicable) and the Introduction, and should be 150–200 words long. The aim should be to make your findings accessible to a wide audience that includes both scientists and non-scientists. Sample summaries can be found on our website under Submission Guidelines: 

https://journals.plos.org/globalpublichealth/s/submission-guidelines#loc-parts-of-a-submission

Additional Editor Comments (if provided):

Authors are requested to address the comments from the reviewers.

Reviewers' comments:

Reviewer's Responses to Questions

**Comments to the Author**

1. Does this manuscript meet PLOS Global Public Health’s publication criteria ? Is the manuscript technically sound, and do the data support the conclusions? The manuscript must describe methodologically and ethically rigorous research with conclusions that are appropriately drawn based on the data presented.

Reviewer #1: Yes

Reviewer #2: Yes

2. Has the statistical analysis been performed appropriately and rigorously?

Reviewer #1: N/A

Reviewer #2: N/A

3. Have the authors made all data underlying the findings in their manuscript fully available (please refer to the Data Availability Statement at the start of the manuscript PDF file)?

Reviewer #1: Yes

Reviewer #2: Yes

4. Is the manuscript presented in an intelligible fashion and written in standard English?

Reviewer #1: Yes

Reviewer #2: Yes

5. Review Comments to the Author

Reviewer #1: The study on surgical antibiotic prophylaxis stewardship is both timely and well-written overall. However, there appears to be a gap in clearly articulating the factors influencing health workers' uptake of the stewardship programme.

To improve the paper's clarity and impact, please consider the following request:

Could you explicitly list and discuss the key factors that were found to influence health workers' uptake of the surgical antibiotic prophylaxis stewardship programme? It would be helpful to:

1. Clearly enumerate these factors, perhaps in a dedicated section or a table.

2. Provide a brief description of each factor.

3. Discuss how each factor positively or negatively impacts the uptake of the stewardship programme.

4. If possible, rank or prioritize these factors based on their perceived importance or impact.

Reviewer #2: Title of the study:

A qualitative study on factors influencing health workers’ uptake of a pilot surgical antibiotic prophylaxis stewardship programme in selected Georgian hospitals

General Comments:

The manuscript investigates the behavioral determinants influencing health workers' uptake of a pilot surgical antibiotic prophylaxis (SAP) stewardship program in selected Georgian hospitals. The manuscript is well-written, with an explicit methodology and comprehensive results. However, there are several points that require revision:

Introduction

• Lines 61–63: Recent data on antimicrobial resistance (AMR) should be incorporated to enhance the context specificity of the introduction. For instance, the following reference can be considered: https://doi.org/10.1016/S0140-6736(24)01867-1.

• Lines 68–70: The sentence, “Addressing surgical antibiotic prophylaxis (SAP) requires an appreciation and understanding of the behavioural and cultural context influencing health-care professionals’ practices and decisions,” could be revised. The term "addressing" may not be appropriate here as SAP is a type of antibiotic indication rather than a problem. Consider rephrasing this for clarity and precision.

• Lines 80–82: The authors discuss the inappropriate use of antibiotics, including in surgical prophylaxis, but there is no reference specifically addressing inappropriate use in surgical prophylaxis. If available, such a reference should be included to strengthen the argument.

Methods

• The methodology is well-articulated and thorough.

• Figure 1: The COM-B model presented in the figure could be tailored more specifically to ASP rather than focusing broadly on AMR and ABR. This adjustment would enhance its relevance to the study.

• Lines 155–156: The sentence, “Given that only one participant in each target group was engaged in the ASP at each hospital, data was almost entirely collected via semi-structured, in-depth interviews (IDIs),” could be clearer. Currently, it suggests that only one individual from each target group per hospital was involved in the ASP. Rewriting this for clarity would improve reader understanding.

Results

• The findings from the secondary target group (influencers) are not discussed. These participants likely have relevant perspectives or interests regarding the ASP program. Including their views would provide a more comprehensive analysis of the program's impact.

Discussion

• Line 368: If “Barriers to health worker ASP uptake related to the ASP delivery and health system context” is a heading, it should be formatted in bold. If it is part of the text, the sentence is incomplete as written.

• Lines 371–373: The discussion mentions the surgical site infection (SSI) reporting and feedback patterns of hospitals where the ASP was not introduced. If data on the reporting patterns of hospitals where the ASP was introduced is available, adding a comparative analysis would strengthen the discussion and provide valuable insights.

6. PLOS authors have the option to publish the peer review history of their article (what does this mean? ). If published, this will include your full peer review and any attached files.

**Do you want your identity to be public for this peer review?** For information about this choice, including consent withdrawal, please see our Privacy Policy .

Reviewer #1: **Yes: ** Bashar Haruna Gulumbe

Reviewer #2: No

---

## [Decision Letter · Decision Letter 1]

7 Jan 2025

PGPH-D-24-01470R1

A qualitative study on factors influencing health workers’ uptake of a pilot surgical antibiotic prophylaxis stewardship programme in selected Georgian hospitals

Dear Dr. Warsi,

Thank you for submitting your manuscript to PLOS Global Public Health. After careful consideration, we feel that it has merit but does not fully meet PLOS Global Public Health’s publication criteria as it currently stands. Therefore, we invite you to submit a revised version of the manuscript that addresses the points raised during the review process.

We look forward to receiving your revised manuscript.

Kind regards,

Bipin Adhikari, MBBS, DTM&H, MCTM, MPH, DPhil

Academic Editor

Journal Requirements:

Additional Editor Comments (if provided):

You have used COM-B behavioural model which is fine, but for readers, it would be important to inform #1 that there are several behavioural models, add a line on how they present similarities and differences; and #2 add rationale behind using this particular model.

Page 15, Line 293 to 310

I see the challenge arising when patients (or their family members) take OTC antimicrobials. This is quite inductive based on your data and I recommend authors to discuss this challenge in the discussion, address why people take OTC antimicrobials, compare the Georgian context with the literature that may explain. This could be outside the remit of intended receipients of training but still likely to affect the intervention. Thus this warrants a discussion in few lines.

Reviewers' comments:

Reviewer's Responses to Questions

**Comments to the Author**

1. If the authors have adequately addressed your comments raised in a previous round of review and you feel that this manuscript is now acceptable for publication, you may indicate that here to bypass the “Comments to the Author” section, enter your conflict of interest statement in the “Confidential to Editor” section, and submit your "Accept" recommendation.

Reviewer #1: All comments have been addressed

Reviewer #2: All comments have been addressed

2. Does this manuscript meet PLOS Global Public Health’s publication criteria ? Is the manuscript technically sound, and do the data support the conclusions? The manuscript must describe methodologically and ethically rigorous research with conclusions that are appropriately drawn based on the data presented.

Reviewer #1: Yes

Reviewer #2: Yes

3. Has the statistical analysis been performed appropriately and rigorously?

Reviewer #1: N/A

Reviewer #2: N/A

4. Have the authors made all data underlying the findings in their manuscript fully available (please refer to the Data Availability Statement at the start of the manuscript PDF file)?

Reviewer #1: Yes

Reviewer #2: Yes

5. Is the manuscript presented in an intelligible fashion and written in standard English?

Reviewer #1: Yes

Reviewer #2: Yes

6. Review Comments to the Author

Reviewer #1: (No Response)

Reviewer #2: The authors have adequately addressed all my comments. The final decision on the acceptance of this manuscript rests with the editor.

7. PLOS authors have the option to publish the peer review history of their article (what does this mean? ). If published, this will include your full peer review and any attached files.

**Do you want your identity to be public for this peer review?** For information about this choice, including consent withdrawal, please see our Privacy Policy .

Reviewer #1: **Yes: ** Bashar Haruna Gulumbe

Reviewer #2: No

---

## [Editor Report · Decision Letter 2]

3 Mar 2025

A qualitative study on factors influencing health workers’ uptake of a pilot surgical antibiotic prophylaxis stewardship programme in selected Georgian hospitals

PGPH-D-24-01470R2

Dear Dr Warsi,

We are pleased to inform you that your manuscript 'A qualitative study on factors influencing health workers’ uptake of a pilot surgical antibiotic prophylaxis stewardship programme in selected Georgian hospitals' has been provisionally accepted for publication in PLOS Global Public Health.

Best regards,

Bipin Adhikari, MBBS, DTM&H, MCTM, MPH, DPhil

Academic Editor